# Laser-Assisted In Situ Keratomileusis (LASIK) Enhancement for Residual Refractive Error after Primary LASIK

**DOI:** 10.3390/jcm11164832

**Published:** 2022-08-18

**Authors:** Majid Moshirfar, Noor F. Basharat, Nour Bundogji, Emilie L. Ungricht, Ines M. Darquea, Matthew E. Conley, Yasmyne C. Ronquillo, Phillip C. Hoopes

**Affiliations:** 1Hoopes Vision Research Center, Hoopes Vision, Draper, UT 84020, USA; 2John A. Moran Eye Center, Department of Ophthalmology and Visual Sciences, University of Utah School of Medicine, Salt Lake City, UT 84132, USA; 3Utah Lions Eye Bank, Murray, UT 84107, USA; 4University of Arizona College of Medicine—Phoenix, Phoenix, AZ 85004, USA; 5University of Utah School of Medicine, Salt Lake City, UT 84132, USA

**Keywords:** laser-assisted in situ keratomileusis (LASIK), refractive surgery, enhancement, retreatment, refractive error

## Abstract

Background: To evaluate the safety, efficacy, and predictability of laser-assisted in situ keratomileusis (LASIK) enhancement after primary LASIK and compare to Food and Drug Administration (FDA) criteria. Methods: Patients who underwent LASIK enhancement after primary LASIK between 2002 and 2019 were compared to those who underwent LASIK without retreatment. Patient demographics, preoperative characteristics, visual outcomes, and postoperative complications were compared between groups. Epithelial ingrowth (EI) development was stratified based on duration between primary and secondary procedures. Results: We compared 901 eyes with LASIK enhancement to 1127 eyes without retreatment. Age, sex, surgical eye, sphere, cylinder, and spherical equivalent (SE) were significantly different between groups (*p* < 0.05). At 12 months post-enhancement, 86% of the eyes had an uncorrected distance visual acuity of 20/20 or better and 93% of eyes were within ±0.50 D of the target. Development of EI (6.1%) demonstrated an odds ratio of 16.3 in the long-term compared to the short-term (95% CI: 5.9 to 45.18; *p* < 0.0001). Conclusions: Older age at primary LASIK, female sex, right eye, and larger sphere, cylinder and SE were risk factors for enhancement. Risk of EI significantly increased when duration between primary and enhancement procedures exceeded five years. LASIK enhancements produce favorable outcomes and meet FDA benchmarks for safety, efficacy, and predictability.

## 1. Introduction

Uncorrected refractive error is the leading cause of visual impairment worldwide, leading to decreased quality of life, lost educational and job opportunities, and increased morbidity [1,2]. Treatment options available include glasses, contact lenses, and laser refractive surgery [3]. Laser-assisted in situ keratomileusis (LASIK), involving the use of a stromal flap, is currently the most popular refractive procedure [4,5]. LASIK provides many advantages, including faster visual recovery, decreased postoperative pain, less stromal haze, and a shorter duration of medication use postoperatively [5]. Though LASIK is a safe and effective procedure; patients can have residual refractive error afterwards [6,7]. Causes of residual refractive error include under correction, overcorrection, and astigmatism [7]. Regression refers to the gradual shift of the eye to its original preoperative refraction [8] and has been reported to occur anywhere from three months to three years after surgery [8,9,10].

Proposed etiologies for residual refractive error after primary LASIK treatment include high preoperative refractive or astigmatic value, postoperative epithelial hyperplasia, or decreased flap thickness [7]. Various retreatment techniques can be used to manage these residual errors, including repeat LASIK, radial keratotomy, surface ablation, recutting the flap, vertical side cuts, and collagen crosslinking [7]. One study reported that 85% of enhancement procedures were performed within one year of the primary LASIK, and the one-year incidence of retreatment was 10.5% [11]. However, LASIK has not yet been approved as a retreatment technique by the United States Food and Drug Administration (US FDA) and is considered an off-label procedure [12]. Because many retreatment modalities are available, this study seeks to analyze the safety, efficacy, and predictability of LASIK enhancement after primary LASIK and compare it to the criteria established by the FDA.

## 2. Materials and Methods

This is a retrospective chart review of patients who underwent LASIK enhancement after primary LASIK refractive surgery between August 2002 and April 2019 at the Hoopes Durrie Rivera (HDR) research center in Draper, Utah. LASIK enhancement surgeries were performed by two surgeons (MM and PCH). We randomly selected a control group of patients who had their primary LASIK surgery done during this timeframe by the same two surgeons (MM and PCH) at this research center with no subsequent retreatments. Patients with pre-existing ocular pathologies such as age-related macular degeneration, keratoconus, prior retinal tears, cataract development, glaucoma, or abnormal topographies were excluded from this study.

Demographic information, surgical eye, preoperative sphere, preoperative cylinder, preoperative spherical equivalent (SE), and corrected distance visual acuity (CDVA) were compared between the enhancement and non-enhancement groups prior to the primary LASIK surgery. Mann-Whitney U tests were used to evaluate age at the time of the primary procedure, preoperative sphere, preoperative cylinder, preoperative SE, and preoperative CDVA. SE, measured in diopters (D), was calculated by adding half of the cylinder value to the sphere value. CDVA was measured using the Snellen chart, and the values were converted to logMAR for analysis. Chi-squared were performed to analyze sex and surgical eye. Eyes with epithelial ingrowth (EI) were stratified based on duration between primary and secondary procedures and were included in either a short-term (within five years of the primary LASIK) or long-term (at least five years after the primary LASIK) enhancement group. The odds ratio (OR) was calculated using the short-term group as the non-exposure group. *p*-values less than 0.05 were considered statistically significant. Statistical analyses were performed using the GraphPad Prism software version 9.3.1 (GraphPad Software, San Diego, CA, USA).

The data collected from the LASIK enhancement group’s three-month and one-year follow-up intervals were evaluated using the Standard 9 Graphs to assess the outcomes of safety, efficacy, stability, and predictability of the LASIK enhancement procedure [13,14]. The Standard 9 Graphs were plotted using Microsoft Excel (Microsoft Corporation, Seattle, WA, USA).

Efficacy of UDVA outcomes compares preoperative CDVA and postoperative UDVA. Safety is defined in terms of the change in Snellen lines of CDVA where the percentage of eyes that lose two or more lines is significant. The loss of one line is within normal biological variability. Predictability is defined as the correlation between the attempted and achieved SE. For the scatter plots, linear regression analysis with the regression equation, trend line and coefficient of determination (r^2^) were calculated. R^2^ denotes the strength of the correlation between attempted and achieved SE change. Its value may be between zero and one, with a stronger correlation (less scatter) when values are closer to one [13]. The stability plot is the change in refraction in the context of the different time points reported.

Informed consent was obtained from each patient for the procedures and for the use of de-identified clinical data in research. The study and consent procedure were approved by the Hoopes Vision Ethics Committee and adhered to the tenets of the Declaration of Helsinki. The Biomedical Research Alliance of New York (BRANY) Institutional Review Board (protocol code: #A20-12-547-823) approved this retrospective study.

### 2.1. Primary LASIK

The primary LASIK procedures in the enhancement and non-enhancement groups were performed by two surgeons at Hoopes Vision (MM and PCH) and other facilities. For the primary LASIK procedures carried out at Hoopes Vision, the AMO iFS (Abbott Medical Optics, Inc., Santa Ana, CA, USA) femtosecond laser system was used to create the flap. The WaveLight EX500 excimer laser system (Alcon Laboratories, Inc., Fort Worth, TX, USA) was used for stromal ablation with a 6.0 to 6.5 mm central optical zone and 8.5 to 9.0 mm transition zone. The WaveLight laser uses Surgivision’s DataLink software to determine customized treatments. The flap diameter was between 8.5 and 9.0 mm, and the flap thickness was between 100 and 115 µm with the creation of a superior hinge. The postoperative treatment protocol included ofloxacin 0.3% or moxifloxacin 0.5% four times per day for one week. Patients were instructed to apply prednisolone acetate 1% every hour while awake for the first 24 h. After the first 24 h, the prednisolone acetate was decreased to four times per day for one week.

### 2.2. LASIK Enhancement

Two surgeons from HDR (MM and PCH) performed the LASIK enhancements. For the enhancement, the patient was examined at the slit lamp, and the flap edge was lifted approximately half a clock hour with a Sinskey hook. The patient was then taken to the refractive suite where the flap was further lifted with a Seibel LASIK flap lifter/spatula (Ambler Surgical, Exton, PA, USA). The WaveLight EX500 excimer laser system was used for stromal ablation with the settings as stated above. A bandage contact lens was placed after the procedure which remained in place for at least four days. Patients were started on a postoperative drop regimen identical to that used with the original procedure.

## 3. Results

This study compared 901 eyes in the LASIK enhancement group to 1127 eyes in the non-enhancement group. Mean time between primary and enhancement LASIK surgeries was 15.33 ± 16.75 months (range, nine days to 11.48 years). Comparison of the enhancement and non-enhancement group’s demographic and clinical information, refraction, SE, and CDVA prior to the primary LASIK are shown in Table 1. The enhancement group included patients who were significantly older at the time of the primary LASIK (*p* < 0.0001). More females underwent retreatments (*p* = 0.0002), and the right eye required more enhancement procedures (*p* = 0.002). Prior to the initial LASIK, patients in the enhancement group were more myopic and had larger cylinder values (*p* < 0.05). However, CDVA was not statistically different between both groups (*p* = 0.314). The most common complications noted within the LASIK enhancement group were EI (6.1%), dryness (5.2%), micro-striae (MS) (2.8%), and corneal haze (2.7%) (Figure 1). Regarding risk of EI based on time interval between procedures, the long-term enhancement group had an odds ratio (OR) of 16.3 compared to the short-term (95% CI: 5.9 to 45.18; *p* < 0.0001).

### 3.1. Efficacy

Prior to the enhancement procedure, 98% of eyes had a CDVA of 20/20 or better, and 100% had a CDVA of 20/40 or better (Figure 2A). At the three-month follow-up interval, 87% of eyes had a UDVA of 20/20 or better while 99% were 20/40 or better. At the 12-month follow-up interval, 86% of the eyes had a UDVA of 20/20 or better, and 99% had a UDVA of 20/40 or better.

### 3.2. Safety

After three months, the uncorrected distance visual acuity (UDVA) was the same or better than the preoperative CDVA in 82% of eyes and within one line of the CDVA in 95% of eyes (Figure 2B). After 12 months, the UDVA was the same or better than the preoperative CDVA in 83% of eyes and within one line in 96% of eyes. Figure 2C shows that 0.5% of the eyes lost two or more Snellen lines of CDVA after three months and 0.6% lost two or more lines after 12 months.

### 3.3. Predictability

Figure 2D shows a strong relationship between attempted and achieved SE at the three-month visit with an R^2^ value of 0.8401; the absolute SE overcorrection at this timeframe was 0.0837. There was also a strong relationship at the 12-month visit between attempted and achieved SE (R^2^ = 0.8301), and the absolute SE overcorrection was 0.1288. After three months, 95% of the eyes reached an SE within ±0.50 D of the intended target, and 100% reached an SE within ±1.00 D (Figure 2E). After 12 months, 93% and 99% of the eyes were within ±0.50 D and ±1.00 D of the intended target, respectively.

### 3.4. Stability

Figure 2F illustrates the stability of the results postoperatively. Between the one-week and 12-month timepoints, 8% of the eyes had a change in SE that was greater than 0.50 D.

### 3.5. Refractive Astigmatism

Figure 2G shows that 49% of the eyes had astigmatism of up to 0.50 D and 84% had astigmatism of up to 1.00 D preoperatively. At three-months, 95% and 100% of the eyes had astigmatism values of up to 0.50 D and 1.00 D, respectively. At 12 months, 93% had astigmatism up to 0.50 D and 100% had astigmatism of up to 1.00 D. At three-months, mean target induced astigmatism (TIA) was 0.85 ± 0.47 D, mean surgically induced astigmatism (SIA) was 0.87 ± 0.55, and the R^2^ value was 0.1968 (Figure 2H); the absolute overcorrection of the astigmatism vector was 0.1968. At 12-months, mean TIA was 0.88 ± 0.47 D, mean SIA was 0.90 ± 0.54, and R^2^ was 0.2388; the absolute overcorrection of the astigmatism vector was 0.2729. At three months, the angle of error arithmetic mean was 0.4 ± 13.3°, and the absolute mean was 5.9 ± 11.9° (Figure 2I). At 12 months, the angle of error arithmetic mean was 0.4 ± 12.8°, and the absolute mean was 6.5 ± 10.9°.

## 4. Discussion

This study aimed to evaluate the safety, efficacy, and predictability in a large sample of patients undergoing LASIK enhancements after primary LASIK. Older age, the female sex, right surgical eye, and higher initial correction and astigmatism were indications for later retreatment. We found that LASIK enhancement after initial LASIK is safe and effective with good visual outcomes that exceed FDA criteria (Table 2).

Prior studies evaluating LASIK enhancements after primary LASIK have been published (Table 3) [15,16,17,18,19,20,21,22,23,24,25,26,27,28,29,30,31,32,33,34,35,36,37,38,39,40,41,42,43,44,45]. At 12 months, 86% and 99% of our eyes had a UDVA of 20/20 or better and 20/40 or better, respectively. In comparison, 21.1–93.3% of eyes in previous studies had a UDVA of 20/20 or better while 88.2–100% of eyes had a UDVA of 20/40 or better [15,16,17,18,19,20,21,22,23,24,25,26,27,28,29,30,31,32,33,34,35,36,37,38,39,40,41,42,43,44,45]. Our results showed 4% of eyes that lost one line of CDVA and 0.6% of eyes that lost two or more lines at 12 months postoperatively. Existing literature shows 0–28% of eyes that lost one line and 0–31.8% that lost two or more lines of CDVA [15,16,17,18,19,20,21,22,23,24,25,26,27,28,29,30,31,32,33,34,35,36,37,38,39,40,41,42,43,44,45]. At 12 months, there were 93% of eyes within ±0.50 D of the intended target and 99% within ±1.00 D in our study. Other studies show 46.4–100% of eyes within ±0.50 D of the target and 63.4–100% within ±1.00 D [15,16,17,18,19,20,21,22,23,24,25,26,27,28,29,30,31,32,33,34,35,36,37,38,39,40,41,42,43,44,45].

Netto and Wilson conducted the largest study in 2004 evaluating 334 retreated myopic eyes and concluded that relifting the flap provided stable and predictable visual outcomes with few complications, but discouraged flap recutting [19]. They also noted a low complication rate with only 0.4% developing EI after retreatment, which they attributed to surgical technique, specifically lack of blunt dissection prior to relifting the flap. Ortega-Usobiaga et al. launched a large-scale study in 2018 on 3772 myopic and 1424 hyperopic eyes that had flap relifts and compared them to eyes with surface ablation retreatment [39]. No differences in visual outcomes between the groups were observed, and they stated that their safety and predictability outcomes were similar to those found in the literature.

Other studies have established similar risk factors, with our findings of sex and surgical eye being notable exceptions [11,46]. Right eye dominance has been found to be more prevalent than left eye dominance [47]; therefore, it is possible that significantly more patients are receiving LASIK retreatments on the right eye as they are more likely to be right eye dominant, thus noticing more subtle changes in their vision. Lopez-Prats et al. compared patients who had undergone LASIK before becoming pregnant to patients with non-surgically corrected eyes who had become pregnant [48]. They found significantly worsening cylinder and SE in patients with a history of LASIK, and they attributed this to the hormonal changes that occur during gestation. However, Kanellopoulos and Vingopoulos found that pregnancy did not impact the refractive stability after LASIK [49]. It is possible that more females return for retreatment due to refractive changes caused by pregnancy or hormonal changes; however, because of the conflicting literature, we cannot confidently cite these factors as the cause.

The most common complication in our study was EI with a post-enhancement rate of 6.1% (Figure 1). EI is a well-reported postoperative complication of enhancement procedures, with rates ranging from 1.7% to 31% [50,51]. It has been proposed that EI occurs from either epithelial cells implanting during surgery or invading the margin of the LASIK flap [52]. Techniques such as the “Flaporhexis” [53], Nd:YAG laser disruption [54], mechanical debridement, and flap suturing [55] have all been shown as safe and effective treatment options for EI. Perez-Santonja et al. suggested that surgical technique impacts the rate of EI occurrence as the flat spatula may create an irregular epithelial border that prompts EI under the flap edge postoperatively [51]. They recommended various surgical measures that may decrease the rate of EI, including linear epithelial dissection, ample irrigation of the interface, and strong adhesion between the flap edge and stromal bed.

Other studies have reported that a longer duration of time between primary and enhancement procedures leads to an increased rate of EI [39,56]. Upon stratification of our EI complications into short-term and long-term enhancements, the odds of EI were 16-fold greater when the duration between primary LASIK and LASIK enhancement exceeded five years (OR: 16.3, *p* < 0.0001). In 2010, Caster et al. published a study of 3866 eyes with primary LASIK and 646 eyes with flap relifts and found a significantly higher number of eyes with EI in the enhancement group compared to the non-enhancement group [56]. The rate of clinically significant EI was higher when the enhancement was performed three or more years after the primary procedure (7.7% vs 1.0%), but the rate stabilized when the enhancement was performed three to 10 years after the primary LASIK. They suggested that PRK be performed if three or more years have elapsed since the primary LASIK to minimize risk of EI. In 2018, Ortega-Usobiaga compared 5196 eyes with LASIK enhancements to 272 eyes with PRK enhancements after primary LASIK [39]. In the group with flap relifts on myopic eyes, they noted a 13.55% rate of EI. However, when the time interval between primary and secondary procedures reached 20 months, the rate of EI was over 20%. Similar to Caster et al., the authors concluded that PRK or LASIK can be safely performed with short-term enhancements but advised that surgeons consider PRK more strongly with long-term enhancements. Therefore, it is imperative that clinicians screen patients meticulously for better postoperative outcomes.

Despite our large sample size, we acknowledge that our results are limited by the present study’s retrospective nature. Additionally, we did not have keratometry or pachymetry data to determine whether curvature or thickness of the cornea are risk factors for later enhancements. One study noted that moderately myopic eyes with flatter corneas undergoing LASIK have better postoperative visual outcomes compared to moderately myopic eyes with steeper corneas [57]. Although this study did not assess enhancement procedures, we question whether steeper corneas may be a risk factor for later enhancements if they have worse visual outcomes after the primary procedure. However, another study found that preoperative keratometry values in hyperopic eyes did not have any correlation with postoperative visual outcomes after LASIK [58]. Unfortunately, tomography devices were variable between patients in our study, and patients had tomography carried out at varying time intervals. Because our objectives were to compare our results to FDA criteria and prior studies as well as to determine risk factors for enhancements, we did not assess tomography measurements due to this heterogeneity. Further studies are warranted to determine whether differences in tomography are risk factors for retreatment and whether that varies based on myopia and hyperopia. Furthermore, Djoeyre et al. established that eyes undergoing LASIK with a central cornea thickness of less than 400 μm produces good visual outcomes but did not compare these outcomes to eyes with thicker corneas [59]. We also did not have complication rates from our non-enhancement group, so we were not able to compare such rates between primary and secondary procedures. As a result, we cannot comment on whether the stated complications were exacerbations of existing ocular conditions as opposed to novel ones. Caster et al. found that none of their primary LASIK cases (0%) but 15 enhancement cases (2.3%) developed clinically significant EI (*p* < 0.0001), proving that flap relifts significantly increase the risk of EI postoperatively [56].

Older age, female sex, right surgical eye, higher degree of myopia, and astigmatism at the time of the primary LASIK procedure were risk factors for enhancements in the present study. The most common complications after the flap relift were EI, dryness, MS, and corneal haze. We conclude that LASIK enhancements after primary LASIK provides favorable outcomes. However, we suggest that surgeons opt for enhancement techniques other than flap relifts when adequate time has elapsed since the primary LASIK. Although it has not yet been approved as a retreatment technique, our results for LASIK enhancements meet the FDA criteria for safety, efficacy, stability, and predictability at both the three and 12-month follow-up visits.

## Figures and Tables

**Figure 1 jcm-11-04832-f001:**
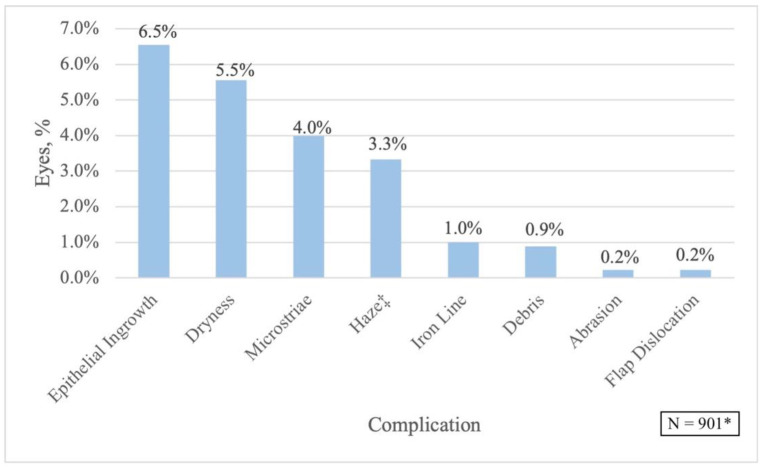
Post-LASIK Enhancement Complications. * Percentages are based on the entire LASIK enhancement group; ^‡^ 2.67% of these were grade 1 haze, 0.67% of these were grade 2 haze.

**Figure 2 jcm-11-04832-f002:**
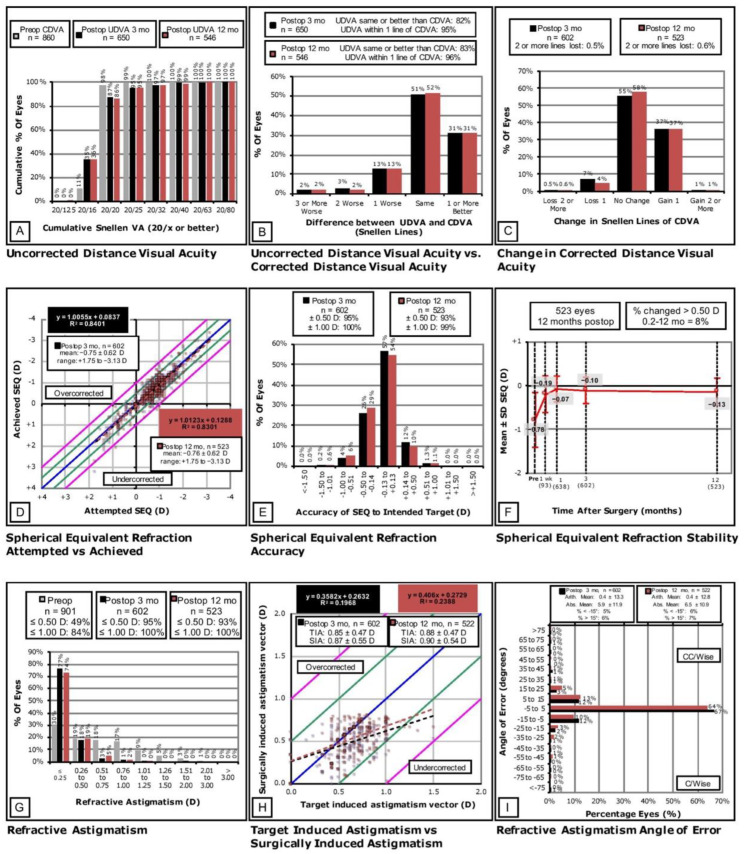
Standard 9 Graphs for the 3- and 12-Month Postoperative Refractive Outcomes after LASIK Enhancement.

**Table 1 jcm-11-04832-t001:** Demographic/Preoperative Comparisons between the Enhancement and Non-Enhancement Groups.

Preoperative Parameters ^a^	LASIK Enhancement after LASIK N = 901	LASIK with No Enhancement N = 1127	*p*-Value
Age, years Mean ± SD Range	37.56 ± 9.051 (22 to 62)	33.88 ± 8.392 (18 to 65)	<0.0001 *
Sex (n, %) Male Female	372 (41.29) 529 (58.71)	558 (49.51) 569 (50.49)	0.0002 *
Surgical Eye (n, %) Right Left	532 (59.05) 369 (40.95)	588 (52.17) 539 (47.83)	0.002 *
CDVA, logMAR Mean ± SD Range	−0.005 ± 0.037 (−0.13 to +0.18)	−0.006 ± 0.034 (−0.13 to +0.30)	0.314
Sphere, D Mean ± SD Range	−3.697 ± 2.831 (−11.50 to +6.00)	−3.242 ± 2.229 (−12.50 to +4.50)	<0.0001 *
Cylinder, D Mean ± SD Range	−1.291 ± 1.238 (−7.25 to 0.00)	−0.900 ± 0.944 (−7.25 to 0.00)	<0.0001 *
SE, D Mean ± SD Range	−4.006 ± 2.849 (−12.13 to +5.75)	−3.692 ± 2.186 (−12.50 to 2.88)	0.004 *

* Statistically significant with a *p*-value < 0.05; ^a^ Prior to the primary LASIK procedure; **Abbreviations:** SD: standard deviation; CDVA: corrected distance visual acuity; logMAR: logarithm of the minimum angle of resolution; D: diopter; SE: spherical equivalent.

**Table 2 jcm-11-04832-t002:** Comparison of Present Study Results to FDA Criteria.

	Parameter	FDA Criteria %	3 mo Post-Enhancement %	12 mo Post-Enhancement %
Safety	Treated eyes with each ocular serious adverse event	<1	0	0
	Loss of at least 2 lines of CDVA	<5	0.5	0.6
	Preoperative CDVA 20/20 or better with postoperative CDVA worse than 20/40	<1	0	0.4
	>2.00 D induced MRC at refractive stability compared to baseline value	<5	0	0
Efficacy	Preoperative CDVA 20/20 or better with postoperative UDVA of 20/40 or better	≥85	99	99
Stability	Change of ≤1.00 D in MRC and MRSE between two refractions postoperatively, either at 1 and 3 months, or over 3 months	≥95	100	100
Predictability	Achievement of MRSE within ±0.50 D of target outcome	≥50	95	93
	Achievement of MRSE within ±1.00 D of target outcome	≥75	100	99

**Abbreviations:** FDA: Food and Drug Administration; mo: month; CDVA: corrected distance visual acuity; MRC: manifest refractive cylinder; UDVA: uncorrected distance visual acuity; D: diopters; MRSE: manifest refraction spherical equivalent.

**Table 3 jcm-11-04832-t003:** Studies in the Literature Evaluating LASIK Enhancement *.

Study	Year	Eyes, n	Interval between Procedures, Months	Mean/Main Follow-Up Interval, Months	UDVA 20/20 or Better, %	UDVA 20/40 or Better, %	CDVA Loss of 1 Line, %	CDVA Loss of ≥2 Line, %	MRSE within ±0.50 D of Intended, %	MRSE within ±1.00 D of Intended, %
Rojas [15]	2002	36	7.83	3	66.6	94.4	28	0	55.6	94.4
Davis ^a^ [16]	2002	164	10	4.8	44.4	98.1	-	-	-	-
		48	10.9	5.63	21.1	78.9	-	-	-	-
Lyle [17]	2003	34	15.5	11.53	37	93	7	0	59	81
Rani [18]	2003	33	6.48	6	-	91	0	0	-	-
Netto [19]	2004	334	8	12	58	92	5	1	80	96
Schwartz [20]	2005	14	10.8	5.3	-	-	0	0	71.4	78.6
Jin ^a^ [21]	2006	53	6.1	6.9	75	-	26	0	91	100
		101	10.1	8.3	75	-	13	0	87	96
Kanellopoulos [22]	2006	22	-	8	-	-	0	0	100	100
Alio ^a^ [23]	2006	44	6.4	12	-	72.6	20.5	31.8	70.5	84.1
		41			-	73.2	19.5	29.2	46.4	63.4
Montague ^b^ [24]	2006	120	-	1	92.3	100	12	1	91	100
				3	88.1	100	5	0	83	100
Alio ^a^ [25]	2006	20	>3	6	-	100	0	0	94.4	100
		20			-	100	0	0	88.8	100
Saeed [26]	2007	60	7.8	22.3	60	95	0	0	77	83
Harter [27]	2007	27	-	12	28.6	100	0	0	57.1	100
Ortega-Usobiaga [28]	2007	86	5.54	5.62	53.49	98.84	17.44	4.65	72.09	96.51
Urbano ^a^ [29]	2008	37	18.07	6	93.3	100	3.3	0	93.6	100
		37			86.7	100	13.3	0	86.7	100
Bragheeth [30]	2008	34	>3	12	43	89	20	0	56	78
Bababeygy [31]	2008	19	23.6	1	55.6	-	15.8	0	55.6	66.7
				3	66.7	-	11.1	0	55.6	88.9
Kashani ^a^ [32]	2009	46	7.8	17.75	86.9	97.8	15.2	0	82.6	95.6
		17	11	14.6	70.6	88.2	7.6	0	88.2	100
McAlinden [33]	2011	60	8.3	6	73.3	-	3.3	0	88.3	98.3
Coskunseven ^c^ [34]	2012	11	18.18	7.72	-	-	0	0	-	-
Santhiago [35]	2012	88	7.3	1 d–12	82	95	2.2	1.1	-	-
Schallhorn [36]	2015	119	14	4	87.4	100	10.1	0.8	87.4	99.2
Frings [37]	2017	113	10.41	1–12	-	-	8	0	78	-
Caster [38]	2018	23	13.9	-	70	100	4	0	85	100
Ortega-Usobiaga ^a^ [39]	2018	3772	14	12 h–3	-	-	-	0.2	81.9	93.6
		1424	17		-	-	-	0.6	70.3	85.2
Alio del Barrio ^a^ [40]	2019	40	12.3y	3	88	100	14	0	97	100
		19			74	100	12	0	76	88
Chan [41]	2020	58	27.4	7.18	91.5	100	17.2	0	-	-
Bamashmus [42]	2020	112	15.5	12	43.8	92.8	1.8	0.9	83.9	94.6
Lee [43]	2020	12	14.5	6.3	92	100	18.2	0	100	100
Hecht [44]	2020	263	29	5.16	49	96	14	1.5	73	90.8
Chang [45]	2022	73	8.6	4.3	-	-	9	0	-	-
Current Study	2022	901	15.33	3	87	99	7	0.5	95	100
				12	86	99	4	0.6	93	99

* Studies from the past 20 years; ^a^ Groups in the study were stratified based on varying parameters; ^b^ One of these eyes had primary PRK; ^c^ Created a second side cut; **Abbreviations:** LASIK: laser-assisted in situ keratomileusis; UDVA: uncorrected distance visual acuity; CDVA: corrected distance visual acuity; MRSE: manifest refraction spherical equivalent; D: diopter; d: day; h: hour; y: year.

## Data Availability

Data sharing is not applicable to this article as no datasets were generated or analyzed during the current study.

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
