# Peer review of "Laser-Assisted In Situ Keratomileusis (LASIK) Enhancement for Residual Refractive Error after Primary LASIK"

_jcm, 2022, doi:10.3390/jcm11164832_

Round 1
Reviewer 1 Report
I read with great interest the manuscript title: Laser-assisted in situ keratomileusis (LASIK) Enhancement for Residual Refractive Error after Primary LASIK. In this study, the authors retrospectively assess the results of enhancement after primary LASIK with respect to safety, efficacy, and predictability.
I find that this is a magnificent article, very well written, understandable and important for our ophthalmological community and especially for those of us who are dedicated to cornea and refractive surgery. The authors study a subject that is rarely treated in the literature but that we nevertheless find in our daily clinical practice. Therefore, I would like to congratulate the authors for their work. In addition, the authors very well collect its limitations (retrospective study) and its strengths (in addition to the important issue addressed, the authors provide us with a large sample size). I have no comments to suggest. I would like to congratulate the authors for their effort and dedication to improve all of us
Reviewer 2 Report
Reviewer compliments authors to conduct this useful study on the outcome of enhancement or retreatment procedure in cases of post Lasik residual refractive error or emergence of refractive error . Despite acknowledged constraints of study in respect of retrospective analysis as well as non availability corneal topography both anterior and posterior and central corneal thickness are undoubtedly compromise out come of the study . In reviewer's view, best results following LASIK with least probability of recurrence or residual refractive error have been observed with flatter corneas of more than 44 diopter and less than 46 diopter as well as pre Lasik corneal thickness of more than 500 micron and post Lasik residual corneal Thickness more than 450 micron are better indices in cases of myopia .Myopia ranging between 3 to 7 diopter along with above mentioned indices have been found of best refractive outcome. On the other end hyperopic corneas are invariably steeper cornea and post Lasik refractive status is always debatable .Hence such cases require enhancement procedure. Reviewer agrees with the authors that re-lifting of Lasik flap should not be the first choice .It is better to have surface ablation in these cases . MMC modulation following lifting flap and post enhancement laser application minimizes risk of EI. To conclude ,best outcome following any refractive laser procedure strictly lies upon meticulous case selection without compromising recommended guidelines .
Reviewer 3 Report
This study retrospectively analyzed the results of lasik enhancement in a large sample size and showed that 99% of refractive errors were within 1D at 12 months postoperatively with LASIK enhancement. Refractive surgery can improve quality of life, but it can also cause irreversible damage to the cornea. Therefore, it is important to evaluate the efficacy and safety of LASIK. Although this study contributes to the safety of refractive surgery, there are several concerns in the methodology section.
1. In this study, the authors evaluate LASIK enhancement comparison with non LASIK enhancement. This study states in the introduction that the LASIK enhancement was performed in accordance with FDA standards, but the standards are not described in the text. Reference 12 was not accessible on my PC because the link was not correct.
2. Line 43. LASIK has some causes of residual refractive errors. The meaning is differenct between regression and undercorrection. Is the contrast between overcorrection and undercorrection?
3. Line 165. Is safety an indicator of visual acuity? Usually, adverse events, etc. are written.
4. Figure 2 provides many informations, however, too busy to read each graph. At least the authors had better explain the purple and red dots, and blue, green, pink solid lines, and black dashed line respectively. I think, purple dots indicate the data of “Postop 3M”. However, I do not understand each line.
I think Figure 2F shows the postoperative refractive changes. However, why is the number of patients at each measurement point different? Please specify the number of pre. The number of people varies by graph. An explanation is needed as to why the numbers are changing.
5. Line 176. If you have done a vector analysis of astigmatism, please specify the j0 and j45 components.
6. Table 2. This comment is partly overlapping with the first one, but what is the basis for the FDA criteria and the parameters that determined the % of 3mo opst enhancement and 12mo enhancement? Do the FDA criteria refer to first LASIK? Furthermore, it is unclear what the refraction value was measured with.
7. This is a serious question, what is the significance of LASIK for early presbyopic eyes, aiming for 0 D? I think the patient will become hypopia in a few years and suffer from eye strain.
Reviewer 4 Report
This study aimed to evaluate the safety, efficacy and predictability of LASIK enhancement. Although the topic is not novel and has been reported multiple times since the early 2000s, it might be a merit in this study due to the sample size. They authors are invited to consider the following recommendations to further strengthen their work:
1. In the interval 2002-2019, 901 eyes required LASIK enhancement vs 1127 not needing, this means that almost 50% of the primary LASIK cases needed an enhancement which is a quite large number. How do the authors explain that ?
2. How would the authors explain that the right eye was found to be a risk factor for LASIK enhancement ?
3. The authors found that female sex is a risk factor for LASIK enhancement. They should discuss the article ''Does Pregnancy Affect Refractive and Corneal Stability or Corneal Epithelial Remodeling After Myopic LASIK?'' by Kanellopoulos et al on possible effects of pregnancy on LASIK stability.
4.Missing corneal thickness and keratometry data in this study is a major omission by the authors giver the key role of both when screening candidates and planning LASIK.
5. Given no thickness and keratometry data, how did the authors made sure to exclude forme fruste keratoconus?
6. Was the surgical techique identical between all 4 surgeons ? please add their initials.
7. Did the re-treatmetn exacaerbated eye dryness experiecned after the primary LASIK ?
Round 2
Reviewer 3 Report
The authors' intention and mine seem to be misaligned. This paper does not state what was analyzed in the methodology. This might be acceptable in a specialized refractive surgery journal such as JCRS. However, this is a general journal, and care should be taken not to confuse the reader. I recommend that the methodology be carefully written so that many people can see the authors' work and become interested in refractive surgery.
Thank you for your valuable comments. Our primary objective is to evaluate whether LASIK enhancement, which is currently an off-label procedure, can pass the FDA requirements in the same manner that the original LASIK surgery was evaluated in terms of safety, efficacy, and predictability. The FDA guidelines for safety, efficacy, and predictability have been in place for the primary LASIK surgery and are included in reference 12.
1. Reference 12 is still not displayed. What is this link? The reader cannot be sure that the link is incorrect, although the text does not state the FDA's criteria.
Thank you for your valuable comment. Safety is related to the postoperative UDVA compared to the preoperative CDVA and the loss of Snellen lines of CDVA. We reported the safety outcomes in the Standard 9 Graphs as well as in the text from lines 152-158. We are not sure if line 165 corresponds to the same thing.
2. Although I did not communicate it well ... . Please write the metrics in the method section. I do not understand if you suddenly write the evaluation index in the results section.
Thank you for your question. We tried to abide by the Standard 9 Graphs that many journals prefer to use. Therefore, we are using those graphs to reflect the visual outcomes. The black dots are used for the 3 month postoperative outcomes, and the black dashed lines correspond to the line of best fit for those values. The red dots are used for the 12 month postoperative outcomes, and the red dashed lines correspond to the line of best fit for those values. The blue line is the target value, the green lines show the points within 0.5 D of the target, and the purple lines show the points within 1.0 D of the target. The number of eyes vary because every eye does not have the corresponding measurement at each time interval. The graphs incorporate all available data for each measurement at each time point.
3. Did the authors search for "Standard 9 Graphs" on Google or elsewhere? In the overrated google search, the JCRS graph has 79 citations. The number of citations in web of science is 57. Thus, these graphs are by no means standard, although JCRS defines them. Only JCRS subscribers can tell everything from the titles of the graphs. As you have explained to me, I recommend you add explanatory text to the graphs as well.
Thank you for your comment. We performed astigmatic evaluation adopted by the Standard 9 Graphs published by Waring et al in figures 2G and 2H. This analysis is not based on the Alpins methods.
4. Usually, vector analysis in astigmatism has means of J0 and J45. This paper does not include any analysis methods in the Methods section, although the results suddenly show the parameters. Again, nine graphs are not a universal standard. Please clarify your analysis method: section 3.5 states R2 value, although is this a linear regression analysis? Is this a linear regression analysis, or do you use another method to fill in missing values?
Thank you for your valuable comment. As we all know, LASIK enhancement is an off-label procedure. The authors wanted to know whether LASIK enhancement could pass the same FDA guidelines that the primary LASIK surgery underwent in terms of safety, efficacy, and predictability. We wanted to use the same guidelines for the enhancement procedure to prove that enhancement can be approved. The guidelines are outlined in Table 2 of our study.
5. I fully understand the intention of the authors. Although as mentioned above, the evaluation indicator is unclear just by looking at the method section. Could you cite Table 2 in Line 86, or prepare a separate table of evaluation indices?
Thank you for your valuable clinical intuition. We also believe that presbyopic middle aged patients should not undergo LASIK surgery because they will lose their reading capability as they become more presbyopic.
6. Thank you!
